# COVID-19 Follow-App. Mobile App-Based Monitoring of COVID-19 Patients after Hospital Discharge: A Single-Center, Open-Label, Randomized Clinical Trial

**DOI:** 10.3390/jpm12010024

**Published:** 2022-01-01

**Authors:** Ester Marquez-Algaba, Marc Sanchez, Maria Baladas, Claudia España, Hermes Salvatore Dallo, Manuel Requena, Ariadna Torrella, Bibiana Planas, Berta Raventos, Carlos Molina, Marc Ribo, Benito Almirante, Oscar Len

**Affiliations:** 1Department of Infectious Diseases, Vall d’Hebron Institut de Recerca (VHIR), Hospital Universitari Vall d’Hebron, Vall d’Hebron Barcelona Hospital Campus, Passeig Vall d’Hebron 119-129, 08035 Barcelona, Spain; marcsanchezbenito@gmail.com (M.S.); aritorrella@gmail.com (A.T.); bplanasribas@gmail.com (B.P.); berta.raventos@vhir.org (B.R.); balmirante@vhebron.net (B.A.); olen@vhir.org (O.L.); 2Stroke Unit, Department of Neurology, Vall d’Hebron Institut de Recerca (VHIR), Hospital Universitari Vall d’Hebron, Vall d’Hebron Barcelona Hospital Campus, Passeig Vall d’Hebron 119-129, 08035 Barcelona, Spain; ictus.to@gmail.com (M.B.); m.requenaruiz@gmail.com (M.R.); cmolina@vhebron.net (C.M.); marcriboj@hotmail.com (M.R.); 3School of Medicine, Universitat Autònoma de Barcelona, 08193 Bellaterra, Spain; claudia.esfu@gmail.com (C.E.); salvatrov@gmail.com (H.S.D.); 4Spanish Network for Research in Infectious Diseases (REIPI RD19/0016), Instituto de Salud Carlos III, Avda. de Monforte de Lemos 5, 28029 Madrid, Spain

**Keywords:** COVID-19, SARS-CoV-2, personalized follow-up, pneumonia, app

## Abstract

Introduction: In the midst of a pandemic, apps can be used to provide close follow-up, ensure that patients are monitored at home, avoid excessive pressure on medical facilities, prevent the movement of people (both patients and health professionals), and reduce the risk of infection. Objective: To adapt and validate the use of a smartphone application for outpatient follow-up of COVID-19 patients after hospital discharge. Methods: We conducted an open-label clinical trial at Hospital Universitari Vall d’Hebron in Barcelona, Spain. Patients were randomly assigned in a 1:1 ratio to be followed by the Farmalarm app or by their primary care center. The primary endpoint was the reduction in the need for in-person return visits. Results: From 31 March to 4 May 2020, 150 patients were enrolled in the study at hospital discharge: 74 patients were randomized to the experimental group, and 76 to the control group. All patients in the control group and all except for six in the experimental group completed the study. During hospitalization, before study inclusion, all but 4 (97.3%) had viral pneumonia, 91 (60.7%) required supplemental oxygen, and 16 (10.7%) required intensive care unit (ICU) admission. COVID-19–related return visits to the emergency department were significantly higher in the control group (7.9% vs. 0%; *p* = 0.028) in the per-protocol analysis. Telephone consultations with the emergency department were performed by 12 (15.8%) patients in the control group and 0 (0%) in the experimental group (*p* < 0.001). Satisfaction with outpatient monitoring was rated higher by the experimental group (5 vs. 4 points; *p* < 0.001). Conclusions: Following COVID-19 hospital discharge, home follow-up via a mobile app was effective in reducing in-person return visits without undermining patient satisfaction or perception of health, compared with standard follow-up.

## 1. Introduction

Since its outbreak in December 2019 in Wuhan, China, the severe acute respiratory syndrome coronavirus 2 (SARS-CoV-2) infection rapidly spread, causing more than 200 million infections and almost 5 million deaths to date [1].

The first COVID-19 was diagnosed in Spain on January 31 2020, and just like in the rest of the world, it exponentially increased in months [2]. Thanks to the efforts of clinicians and the scientific community, the clinical characteristics of patients with coronavirus disease (COVID-19) were early described, with most presenting mild symptoms, and progression to SARS in over 25 to 30% of patients with pneumonia [3,4,5,6,7].

It caused a therapeutic challenge for health workers due to the complete absence of evidence about its management, doubts about contagiousness, and uncertainty about patients’ evolution and the potential complications appearing on discharge, inherent in new diseases.

All efforts were prioritized in diagnosing and treating patients with acute COVID-19, both in the hospital and the primary care settings. Ambulatory monitoring of such a high volume of patients was unaffordable at the time, so once patients were discharged, the system was unable to guarantee an adequate follow-up.

Prior to the outbreak of the pandemic, some studies had already described the potential of telemedicine in disasters and public health emergencies [8,9,10]. In the midst of a pandemic, apps can be used to provide close follow-up, ensure that patients are monitored at home, avoid excessive pressure on medical facilities, prevent the movement of people (both patients and health professionals), and reduce the risk of infection. Additionally, it allows maintaining a personalized follow-up, reducing the post-discharge stress and anxiety in the context of uncertainty.

The aim of our study was to adapt and validate the use of Farmalarm, a smartphone application (app) originally developed by the Stroke Unit at Hospital Universitari Vall d’Hebron in Barcelona, Spain [11]. This app allowed close and personalized monitoring of 74 COVID-19 patients after hospital discharge. The system enabled a two-way communication between the outpatients and a multidisciplinary team in order to detect possible complications (signs of pulmonary embolism, reappearance of respiratory failure, among others). Patients were provided with information about rehabilitation exercises, and they were able to consult any doubt.

## 2. Methods

### 2.1. Study Design and Participants

We conducted the COVID-19 Follow-App trial, a single-center, open-label, randomized clinical trial at Hospital Universitari Vall d’Hebron, a 1100-bed teaching hospital in Barcelona, Spain.

Patients were eligible if they met the following inclusion criteria: aged 18 years or older; availability of a mobile device, such as a smartphone or a tablet with internet connectivity; and hospital discharge after admission for COVID-19 diagnosed by real-time polymerase chain reaction (RT-PCR). Patients were excluded if they were health care professionals (physicians or nurses), or had been discharged to health care facilities or medicalized hotels to complete the isolation period.

The presence of SARS-CoV-2 was determined by RT-PCR performed on nasopharyngeal and oropharyngeal swabs. We used commercial Allplex™ 2019-nCoV multiplex RT-PCR (Seegene, Seoul, South Korea) for the detection of three viral targets (E, N, and RNA-dependent RNA polymerase, RdRp), and an internal control [12,13]. Total nucleic acids (DNA/RNA) were extracted from respiratory specimens using NucliSENSeasyMAG (BioMerieux, Craponne, France) and STARMag Universal Cartridge Kit (Seegene, South Korea) according to the manufacturer’s instructions.

Demographic and socioeconomic data were collected, and included age, nationality, cohabitants, educational level (primary, secondary, university), as well as severity-related information (presence of pneumonia, need for supplemental oxygen, intensive care unit (ICU) admission, and immunomodulatory treatment).

The pneumonia diagnosis was supported by diagnostic images consistent with COVID-19, usually chest X-ray or, in some cases, CT scan.

### 2.2. Study Interventions

Farmalarm (Figure 1) is an app designed to increase stroke awareness and therapeutic adherence through visual and audible notifications. The app software offers versatility to modify the parameters to be monitored and the resulting information, which were adapted for our purpose.

Two infectious disease specialists and a software engineer from the Stroke Unit were involved in adapting the original mobile app and the original Farmalarm web platform, implementing the project specifically for COVID-19 post-discharge follow-up.

The web platform (WP) was managed by a health care monitor (hcM) (physician or nurse trained in COVID-19) who was in charge of monitoring alarm notifications and answering chat questions. The project coordinator was an infectious disease specialist in charge of solving complex issues and performing interim visits, if necessary, and all end-of-study visits.

After a pilot study with 10 nonrandomized patients, the project was launched. During the randomization phase, two physicians were in charge of patient recruitment and initial patient interviews. Intervention began once the patient was discharged, and randomized. At that time, patients assigned to the experimental group were given a personal access code ensuring data privacy and were lent an intelligent pulse oximeter (Smart Pulse Oximeter OL-750, LifeVit, Guangdong Biolight Meditech Co., LTD, Zhuhai, China).

In a 15-min interview, the patient was also trained to use the app and pulse oximeter, and to measure vital signs.

During the following 2 weeks, patients recorded their vital signs twice daily. The intelligent pulse oximeter downloaded data directly to the WP each time the patient used it. Patients were also required to answer a symptom survey every day during the same period. Any abnormal vital signs or survey responses were automatically reported to the hcM, who could then contact the patient immediately. Private chat communication with the hcM was available from 8:00 to 17:00. Patients also received information about the disease, isolation recommendations, and rehabilitation exercises, in both video and PDF format.

Patients assigned to the control group received regular follow-up in the primary care setting via telephone calls to monitor patients’ symptoms. Depending on staff availability at each primary care center and health care workloads, these telephone calls were made every other day, weekly, or just once during the follow-up period.

All participants were monitored for a minimum of two weeks, and they answered an end-of-follow-up survey (Table 1) and a Patient-Reported Outcomes Measurement Information System (PROMIS) questionnaire about their overall health [14]. At the end of follow-up, patients were interviewed via app-based videoconferencing (experimental group) or telephone (control group). All patients also answered a Hospital Anxiety and Depression Scale (HADS) on the day of discharge and again at the end of follow-up [15].

### 2.3. Randomization and Masking

Patients who met the enrolment criteria were randomly assigned in a 1:1 ratio to follow-up by with the Farmalarm app or in the usual primary care setting. The randomization list was computer-generated using a simple randomization function with Excel.

### 2.4. Outcomes

The primary outcome of the study was reduction in the need for in-person return visits.

Secondary outcomes were degree of anxiety, satisfaction, and perception of global health at the end of follow-up.

The intention-to-treat (ITT) analysis included all patients who underwent randomization. Any patients lost to follow-up were considered failures in both strategies. The per-protocol (PP) analysis included patients who completed all end-of-follow-up requirements.

### 2.5. Statistical Analysis

Descriptive results are expressed as medians and interquartile ranges (IQR) or means (± standard deviation [SD]) for continuous variables, after confirming normal distribution with the Kolmogorov–Smirnov test, and as frequencies and percentages for categorical variables.

Differences between the two groups were analyzed by the chi-squared test or Fisher’s exact test for categorical variables, and by the Mann–Whitney U test or Student’s *t*-test as indicated for continuous variables. Two-tailed *p* values of less than 0.05 were considered to indicate statistical significance. All analyses were performed with SPSS Statistics, version 23.0 (IBM Corp., Armonk, NY, U.S.A.).

Due to the lack of previous information on the effects of this kind of intervention, we estimated that the rate of return visits by the end of follow-up would be 10% lower in the experimental group (5% vs. 15%). Consequently, we calculated an enrolment of 165 patients for the study to have 80% power, with a P value of 0.05, assuming a withdrawal rate of 15%.

## 3. Results

From 31 March to 4 May 2020, 150 patients were included in the study. During that period, our hospital admitted 957 patients with COVID-19. Around 700 were excluded due to death, inpatient status at the end of the inclusion period, or exclusion criteria. Around 250 patients were assessed to participate in the study, of which 150 were finally enrolled. The remaining were excluded because they were not interested in follow-up with the app, did not have mobile data coverage or a Wi-Fi network at home, or had been discharged before the interview. A total of 74 patients were randomized to the experimental group, and 76 to the control group. All patients in the control group and all except six in the experimental group completed the study. The reasons were loss of usual residence, language barrier and inability to answer the surveys, or technical issues related to the patient’s smartphone or app usage (loss of password, inability to answer surveys, lack of mobile coverage). Data are shown in Figure 2.

There were no significant differences in baseline characteristics between the two groups, as shown in Table 2. A total of 85 (56.6%) patients were men, and the median age was 53 years (range, 23–84). All but 4 (97.3%) had viral pneumonia, 91 (60.7%) needed supplemental oxygen during hospital admission, and 16 (10.7%) required ICU admission. Additionally, 30 (20%) received immunomodulatory therapy, mainly tocilizumab (23), steroid bolus (5), or cyclosporine (1), and 2 were randomized to a blind clinical trial of sarilumab vs. placebo. The median duration of hospitalization was 6 days (IQR, 5 to 10).

According to the PP analysis (Table 3), patients in the control group were significantly more likely to return to the emergency department (ED) for COVID-19–related reasons than those in the experimental group (7.9% vs. 0%; *p* = 0.029). The reasons for the return visit were thoracic pain (three patients), nasal congestion (two patients), and tracheotomy wound infection (one patient). Only one patient in the experimental group was referred to the ED, but the visit was not related to COVID-19 (acute urinary retention). No differences were observed in the intention-to-treat analysis.

None of the patients in the experimental group needed to consult Catalan emergency services by telephone (numbers 061 and 112) or the primary care setting, and all questions were answered via the live chat capabilities of the app. Conversely, 12 (15.8%) patients in the control group called emergency services. The difference was also statistically significant in the PP (*p* < 0.001), but not in the ITT analysis (Table 3).

Satisfaction with outpatient monitoring was rated more highly by the experimental group in both the PP and the ITT analyses. There were no statistically significant differences in the health status questionnaire or anxiety scale by the end of follow-up, as shown in Table 3.

## 4. Discussion

Although originally developed for stroke patients, the Farmalarm app has been shown in our study to be a safe, secure tool for post-discharge home monitoring of patients with COVID-19. Use of the app during a pandemic is of particular interest, as it helps minimize exposure risks in medical facilities, and allows effective follow-up of a high volume of patients, thus streamlining the use of health resources while ensuring patient and professional convenience and satisfaction.

The large influx of patients from March to May 2020 put considerable pressure on the Spanish national public health system and medical professionals, leading us to seek new solutions to the urgent need for efficient, high-quality approaches to patient follow-up.

At the moment our study was developed, the use of telemedicine in public emergencies had already been described [8,9,10]. However, COVID-19 is transforming the telemedicine landscape with enormous speed. Hundreds of platforms, mobile applications, and different telehealth technologies have been developed in few months to support COVID-19 diagnosis, treatment, and home follow-up, and also for monitoring other chronic pathologies [16,17,18,19,20,21,22,23].

Patients who require hospital admission for COVID-19 represent a population of significant volume and special characteristics, due to the severity of the disease and the uncertainty of its progression, as the infection has an unpredictable clinical course, especially in the early stage.

Besides, several complications have been described after the acute phase of COVID-19, such as the appearance of pulmonary tromboembolism and other thrombotic events [24,25], SARS-CoV-2 organizing pneumonia [26], protracted COVID-19 in immunocompromised patients [27], myopericarditis [28], and bacterial superinfection, among others. One study performed in our setting described a readmission cumulative incidence of 5.4% being patients with prior diagnosis of heart failure, length of stay >13 days, treated with corticosteroids, or who developed pulmonary thromboembolism, putting them at higher risk of readmission [29]. For these reasons, it is vital to keep these patients at home and monitored effectively, while also not stretching health resources more than necessary. The COVID-19 Follow-App project was designed to help meet this need.

Our study had several limitations, as it was a single-center study and required patient ability to use an app. Of the six patients excluded, four did not participate due to technical problems related to the app or the internet connection, and one due to a language barrier. Therefore, strict screening criteria can help ensure reliable monitoring with no need for a “safety net” such as regular primary care follow-up. Like many other measures taken during the pandemic, this small-scale project was rapidly set up to ease the heavy burden on clinicians during the pandemic.

Because our study was undertaken during a newly emerging public health crisis, the sample size was calculated on the basis of an estimated return visit rate that was ultimately lower than expected. However, the results showed that the proportion of patients who attended the ED for COVID-19–related reasons was significantly lower in the experimental group.

Interestingly, despite the fact that the control group needed to consult with emergency services after discharge more frequently, satisfaction and perception of health were not significantly different between the groups. In general terms, all patients were satisfied with the quality of care during hospitalization, and were very grateful for the enormous effort made by primary care to perform the control telephone call. According to the questionnaires, the points that patients highlighted as very positive about the app were the feeling of security provided by a personalized follow-up, and the possibility of receiving answers to their doubts almost immediately by an infectious disease specialist. The latter aspect was of vital importance at a time when information reached the population in a massive and unfiltered way through the mass media. We initially hypothesized that the app would lessen any feelings of uncertainty or panic. However, it was not seen to reduce the anxiety score at home compared with the control group, possibly because a generic questionnaire was used, rather than an instrument more suitable to the actual endpoint of the study.

Due to limited knowledge of the clinical course of the disease in that moment, this study was particularly cautious: all patients enrolled had been hospitalized long enough to ensure clinical stability and/or progression toward symptom resolution.

The app has been shown to be reliable for follow-up purposes and, therefore, a useful tool in the future to shorten hospital stays among patients with a respiratory disease. The low material and staff costs required for the project, and the versatility of the Farmalarm app have led to its recent adaptation and implementation in other departments in our hospital, such as the Nephrology, Hematology, and Chronic Heart Failure units.

The COVID-19 pandemic has hastened the arrival and use of telehealth. The challenge now is to promote its development, and bring it up to the highest standards of health care.

## 5. Conclusions

In summary, we have shown that home follow-up after COVID-19 hospital discharge via the Farmalarm app was more effective in reducing in-person return visits without undermining patient satisfaction and perception of health, when compared with standard follow-up.

## Figures and Tables

**Figure 1 jpm-12-00024-f001:**
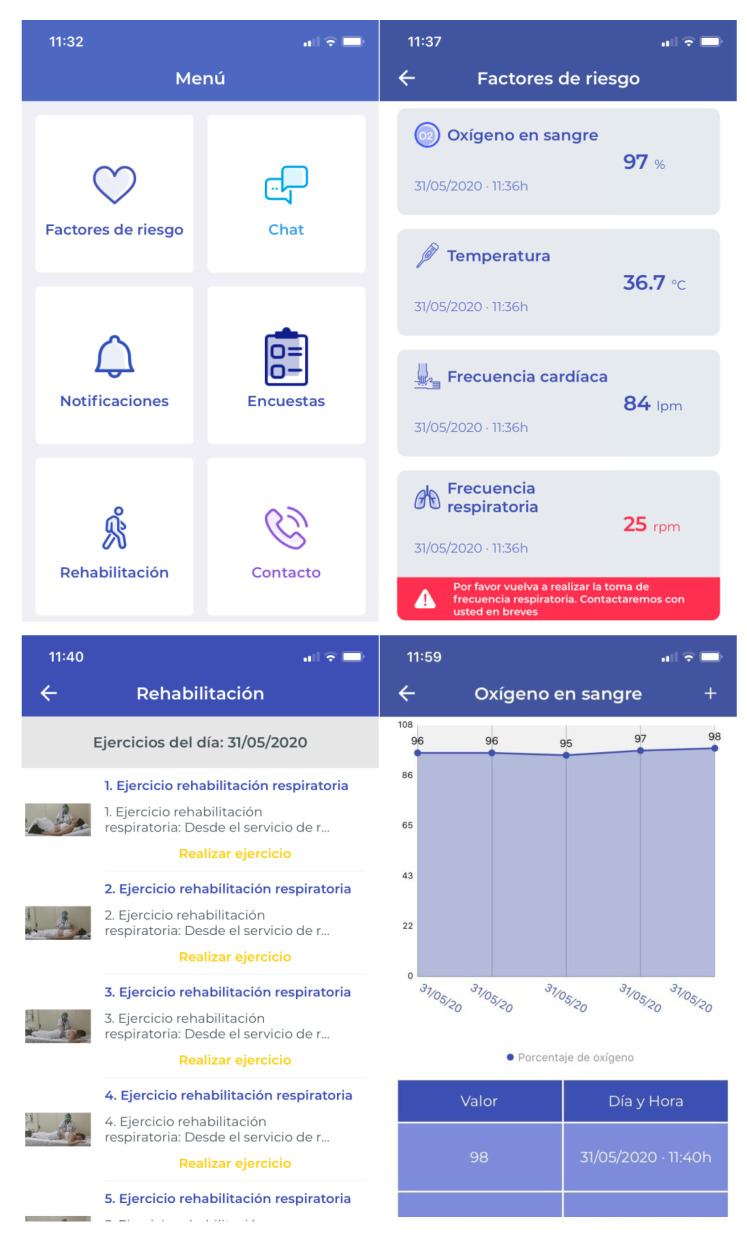
User interface.

**Figure 2 jpm-12-00024-f002:**
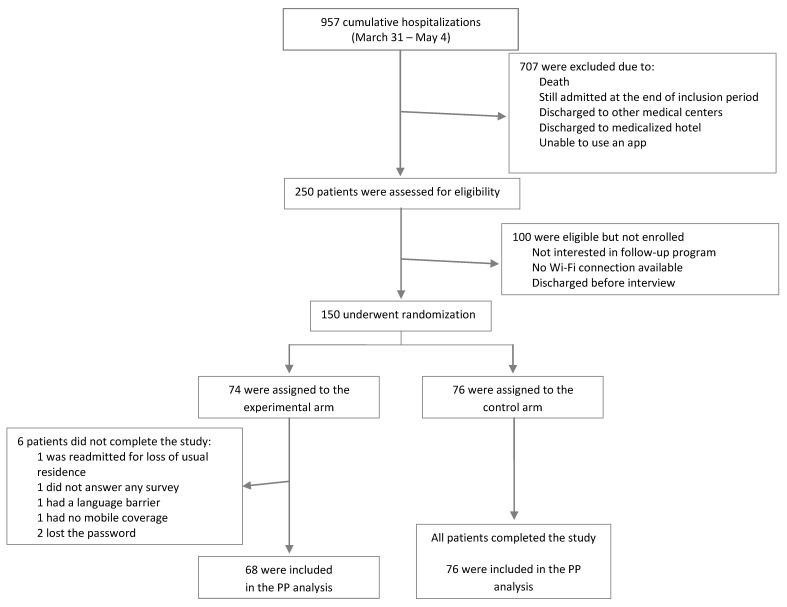
Enrollment and randomization.

**Table 1 jpm-12-00024-t001:** End-of-follow-up questionnaire.

Did you have to contact any other health care services besides us to answer questions on those days?	No = 1 point; Yes = 0 points
Did you have to visit emergency department (hospital or primary care) on those days?	No = 1 point; Yes = 0 points
Have your questions about the disease been answered?	Yes = 1 point; No = 0 points
Have your questions about isolation measures at home been answered?	Yes = 1 point; No = 0 points
Did you feel safe at home after discharge?	Yes = 1 point; No = 0 points
From 0 to 5, how satisfied are you with the care you received on the days after hospitalization?	0 = Extremely dissatisfied; 1 = Moderately dissatisfied; 2 = Slightly dissatisfied; 3 = Neutral; 4 = Moderately satisfied; 5 = Extremely satisfied.

**Table 2 jpm-12-00024-t002:** Demographic and clinical characteristics.

	All Patients; *n* = 150	COVID-19 Follow-App; *n* = 74	Control Group; *n* = 76	*p* Value
Male sex—no. (%)	85 (56.7)	42 (56.8)	43 (56.6)	0.982
—yr †	53.5 (45.7-60)	53.5 (46-59)	53.5 (43.2-63)	0.398
Pneumonia – no. (%)	146 (97.3)	72 (97.3)	74 (97.4)	0.978
Oxygen required—no. (%)	91 (60.7)	45 (60.8)	46 (60.5)	0.972
ICU admission—no. (%)	16 (10.7)	9 (12.2)	7 (9.2)	0.605
IM treatment—no. (%)	30 (20)	15 (20.3)	15 (19.7)	0.548
Tocilizumab	23	14	9	0.154
Steroids	5	2	3	0.670
Sarilumab	2	1	1	0.685
Cyclosporine	1	0	1	0.695
Educational level				
Primary—no. (%)	49 (32.9)	21 (28.4)	28 (37.3)	0.301
Secondary—no. (%)	52 (34.9)	25 (33.8)	27 (36)	0.301
University—no. (%)	48 (32.2)	28 (37.8)	20 (26.7)	0.301

† Age is expressed as median and interquartile range. ICU denotes intensive care unit, and IM denotes immunomodulatory.

**Table 3 jpm-12-00024-t003:** Outcomes according to group.

	Intention-to-Treat Analysis			Per-Protocol Analysis		
	COVID-19 Follow-App *n* = 74	Control Group *n* = 76	*p* Value	COVID-19 Follow-App *n* = 68	Control Group *n* = 76	*p* Value
Number of return ED visits (%) *	6 (8,1)	6 (7.9)	0.962	0	6 (7.9)	0.029
Number of return phone visits (%)	6 (8,1)	12(15.8)	0.148	0	12(15.8)	<0.001
Satisfaction questionnaire, median (IQR)	5 (4–5)	4 (3-5)	<0.001	5 (4-5)	4 (3-5)	<0.001
HADS score, mean (±SD) †	5.5 (±4.4)	5.8 (±4.0)	0.655	5.2 (± 4.1)	6.2 (±4.2)	0.249
PROMIS global health survey, mean (±SD) ∫	12.4 (±3.1)	13.1 (±3.2)	0.180	12.2 (±3.1)	13.1 (±3.2)	0.106

* Need to face-to-face emergency department visits expressed as number and percentage. † HADS score 0–7, normal; 8–10, borderline abnormal (borderline case); 11–21, abnormal (case). ∫ PROMIS score from 4 (poor health) to 20 (excellent health). Data expressed as number and percentages unless otherwise indicated. SD, standard deviation; IQR, interquartile range; HADS, Hospital Anxiety and Depression Scale; PROMIS, Patient-Reported Outcomes Measurement Information System.

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
