# Peer review of "COVID-19 Follow-App. Mobile App-Based Monitoring of COVID-19 Patients after Hospital Discharge: A Single-Center, Open-Label, Randomized Clinical Trial"

_jpm, 2022, doi:10.3390/jpm12010024_

Round 1
Reviewer 1 Report
This Covid-19 Follow-App trial is a single-center, open-label, randomized clinical trial in Barcelona, Spain and demonstrates the usefulness of an App-approach in the follow up of Covid 19 patients. Interestingly, both in-person return visits and return phone visits were 0 in the intervention group. Since patient satisfaction and perception of health were not different between the groups, the quality of standard care might have played a role. This could be discussed in more detail.
Author Response
We agree with the reviewer so we have changed the discussion accordingly. The modified text would be:
Discussion:
Interestingly, despite the fact that the control group needed to consult with emergency services after discharge more frequently, satisfaction and perception of health were not significantly different between the groups. In general terms, all patients were satisfied with the quality of care during hospitalization and were very grateful for the enormous effort made by primary care to perform the control telephone call.
Reviewer 2 Report
This paper presents a solution for Mobile App-Based Monitoring of Covid-19 Patients after the treatment. The paper is well written but has some flaws.
Firstly, the premise of the problem is not very clear.
- How this app helps keeping patients at home? It is a monitoring app after hospital visit, that means patients are already treated and needs less monitoring. Then why we need this app?
- How this app helps in contact tracing?
- Also some studies shows that certain group (age 18-40) of patients develop immunity after exposure to the infection, is there a need to trace those patients?
Secondly, the app was used for stroke patients, as claimed in the paper. COVID and stroke patients are two entirely different groups. stroke patients are not contagious but other group is. So , how we can use the same app?
Finally, after exposure to the COVID infection generally patients are more cautious and careful. So, what is the nature of follow up and why its required is not clear.
Reviewer 3 Report
This study investigates mobile app-based monitoring of covid-19 patients after hospital discharge. I should appreciate the authors' time and patient to come up with some results. However, there are still several problems that deduct from the quality of this manuscript. Below are several comments on this work.
- Please replace the Gmail address with your affiliation email.
- I can not find Figures 1 and 2 mentioned in your manuscript.
- The sample size is insufficient. In addition, the patient satisfaction might vary from one hospital to another.
- Did you classify the patients according to the severity of their symptoms?
- The authors should proofread the English writing to improve the study.
Round 2
Reviewer 2 Report
The authors have addressed my concerns.
The paper can be accepted in current form.
Reviewer 3 Report
No comments.